# The Spectrum of *FANCM* Protein Truncating Variants in European Breast Cancer Cases

**DOI:** 10.3390/cancers12020292

**Published:** 2020-01-26

**Authors:** Gisella Figlioli, Anders Kvist, Emma Tham, Jana Soukupova, Petra Kleiblova, Taru A Muranen, Nadine Andrieu, Jacopo Azzollini, Judith Balmaña, Alicia Barroso, Javier Benítez, Birgitte Bertelsen, Ana Blanco, Bernardo Bonanni, Åke Borg, Joan Brunet, Daniele Calistri, Mariarosaria Calvello, Stepan Chvojka, Laura Cortesi, Esther Darder, Jesús Del Valle, Orland Diez, ENIGMA Consortium, Séverine Eon-Marchais, Florentia Fostira, Francesca Gensini, Claude Houdayer, Marketa Janatova, Johanna I Kiiski, Irene Konstantopoulou, Katerina Kubelka-Sabit, Conxi Lázaro, Fabienne Lesueur, Siranoush Manoukian, Ruta Marcinkute, Ugnius Mickys, Virginie Moncoutier, Aleksander Myszka, Tu Nguyen-Dumont, Finn Cilius Nielsen, Rimvydas Norvilas, Edith Olah, Ana Osorio, Laura Papi, Bernard Peissel, Ana Peixoto, Dijana Plaseska-Karanfilska, Timea Pócza, Maria Rossing, Vilius Rudaitis, Marta Santamariña, Catarina Santos, Snezhana Smichkoska, Melissa C Southey, Dominique Stoppa-Lyonnet, Manuel Teixeira, Therese Törngren, Angela Toss, Miguel Urioste, Ana Vega, Zdenka Vlckova, Drakoulis Yannoukakos, Valentina Zampiga, Zdenek Kleibl, Paolo Radice, Heli Nevanlinna, Hans Ehrencrona, Ramunas Janavicius, Paolo Peterlongo

**Affiliations:** 1Genome Diagnostics Program, IFOM - the FIRC Institute for Molecular Oncology, Milan 20139, Italy; 2Division of Oncology and Pathology, Department of Clinical Sciences Lund, Lund University, Lund SE-22381, Sweden; 3Department of Clinical Genetics, Karolinska University Hospital and Department of Molecular Medicine, Karolinska Institutet, Stockholm 17176, Sweden; 4Institute of Biochemistry and Experimental Oncology, First Faculty of Medicine, Charles University, Prague 12853, Czech Republic; 5Institute of Biology and Medical Genetics, General University Hospital and First Faculty of Medicine, Charles University, Prague 12800, Czech Republic; 6Department of Obstetrics and Gynecology, Helsinki University Hospital and University of Helsinki, HUS, Helsinki 00029, Finland; 7Inserm, U900, Institut Curie, PSL University, Paris F-75005, France; 8Mines ParisTech, Fontainebleau F-77300, France; 9Department of Medical Oncology and Hematology, Unit of Medical Genetics Fondazione, IRCCS Istituto Nazionale dei Tumori, Milan 20133, Italy; 10Hereditary Cancer Group, Vall d’Hebron Institute of Oncology (VHIO), Barcelona 08035, Spain; 11Department of Medical Oncology, University Hospital Vall d´Hebron, Barcelona 08035, Spain; 12Human Genetics Group, Human Cancer Genetics Programme, Spanish National Cancer Research Centre, Madrid 28029, Spain; 13Spanish Network on Rare Diseases (CIBERER), Madrid 28029, Spain; 14Genotyping Unit, CEGEN, Human Cancer Genetics Programme, Spanish National Cancer Research Centre, Madrid 28029, Spain; 15Center for Genomic Medicine, Copenhagen University Hospital, Rigshospitalet, Copenhagen 2100, Denmark; 16Fundación Pública Galega Medicina Xenómica-SERGAS, Santiago de Compostela 15706, Spain; 17Instituto de Investigación Sanitaria de Santiago de Compostela (IDIS), Santiago de Compostela 15706, Spain; 18Centro de Investigación en Red de Enfermedades Raras (CIBERER), Madrid 28029, Spain; 19Division of Cancer Prevention and Genetics, IEO, European Institute of Oncology IRCCS, Milan 20141, Italy; 20Hereditary Cancer Program, Catalan Institute of Oncology, ONCOBELL-IDIBELL-IDIBGI-IGTP, CIBERONC, Barcelona 08908, Spain; 21Biosciences Laboratory, Istituto Scientifico Romagnolo per lo Studio e la Cura dei Tumori (IRST) IRCCS, Meldola 47014, Italy; 22Centre for Medical Genetics and Reproductive Medicine, Gennet, Prague 17000, Czech Republic; 23University Modena Hospital, Modena 41124, Italy; 24Àrea of Molecular and Clinical Genetics, University Hospital Vall d´Hebron, Barcelona 08035, Spain; 25QIMR Berghofer Medical Research Institute, Brisbane, QLD, Australia; 26InRASTES, Molecular Diagnostics Laboratory, National Centre for Scientific Research “Demokritos”, Athens 15310, Greece; 27Department of Experimental and Clinical Biomedical Sciences, University of Florence, Florence 50134, Italy; 28Genetics Department, F76000 and Normandy University, UNIROUEN, Inserm U1245, Normandy Centre for Genomic and Personalized Medicine, Rouen University Hospital, Rouen, France; 29Department of Histopathology and Cytology, Clinical Hospital Acibadem Sistina, Skopje 1000, Republic of North Macedonia; 30Hereditary Cancer Center, Hematology, Oncology and Transfusion Medicine Center, Vilnius University Hospital Santaros Klinikos, Vilnius 08410, Lithuania; 31National Center of Pathology, Vilnius University Hospital Santaros Klinikos, Vilnius 08410, Lithuania; 32Service de Génétique, Institut Curie, Inserm, U830, Paris Descartes University, Paris F-75005, France; 33Division of Clinical Genetics, Department of Laboratory Medicine, Lund University, Lund SE-22100, Sweden; 34Institute of Medical Sciences, University of Rzeszow, Rzeszow 35-310, Poland; 35Precision Medicine, School of Clinical Sciences at Monash Health, Monash University, Clayton 3168, Australia; 36Department of Clinical Pathology, The University of Melbourne, Melbourne 3010, Australia; 37Department of experimental, preventive and clinical medicine, State Research Institute Centre for Innovative Medicine, Vilnius 08410, Lithuania; 38Department of Molecular Genetics, National Institute of Oncology, Budapest 1122, Hungary; 39Department of Genetics, Portuguese Oncology Institute of Porto (IPO Porto), Porto 4200-072, Portugal; 40Research Centre for Genetic Engineering and Biotechnology ‘Georgi D. Efremov’, Macedonian Academy of Sciences and Arts, Skopje 1000, Republic of North Macedonia; 41Department of Gynaecology, Center of Obsterics and Gynaecology, Vilnius University Hospital Santaros Klinikos, Vilnius 08410, Lithuania; 42Medical Faculty, University Clinic of Radiotherapy and Oncology, Ss. Cyril and Methodius University in Skopje, Skopje 1000, Republic of North Macedonia; 43Biomedical Sciences Institute, University of Porto, Porto 4050-313, Portugal; 44Familial Cancer Clinical Unit, Human Cancer Genetics Programme, Spanish National Cancer Research Centre, Madrid 28029, Spain; 45Department of Medical Genetics, GHC Genetics, Prague 11000, Czech Republic; 46Department of Research, Unit of Molecular Bases of Genetic Risk and Genetic Testing, Fondazione IRCCS Istituto Nazionale dei Tumori, Milan 20133, Italy; 47Office for Medical Services, Region Skåne, Department of Clinical Genetics and Pathology, Laboratory Medicine, Lund SE-22100, Sweden

**Keywords:** breast cancer predisposition, breast cancer risk factors, *FANCM* truncating variants, mutation spectrum, PTVs

## Abstract

Germline protein truncating variants (PTVs) in the *FANCM* gene have been associated with a 2–4-fold increased breast cancer risk in case-control studies conducted in different European populations. However, the distribution and the frequency of *FANCM* PTVs in Europe have never been investigated. In the present study, we collected the data of 114 European female breast cancer cases with *FANCM* PTVs ascertained in 20 centers from 13 European countries. We identified 27 different *FANCM* PTVs. The p.Gln1701* PTV is the most common PTV in Northern Europe with a maximum frequency in Finland and a lower relative frequency in Southern Europe. On the contrary, p.Arg1931* seems to be the most common PTV in Southern Europe. We also showed that p.Arg658*, the third most common PTV, is more frequent in Central Europe, and p.Gln498Thrfs*7 is probably a founder variant from Lithuania. Of the 23 rare or unique *FANCM* PTVs, 15 have not been previously reported. We provide here the initial spectrum of *FANCM* PTVs in European breast cancer cases.

## 1. Introduction

*FANCM*, the Fanconi anemia (FA) complementation group M gene (OMIM609644), was originally described as one of the members of the FA molecular pathway [1] that is primarily responsible for the repair of the DNA inter-strand crosslinks through homologous recombination. FA is a recessive DNA repair disease characterized by bone marrow failure, congenital malformations, chromosome fragility, and cancer. Although *FANCM* is part of the core complex in the FA pathway, accumulating evidence indicates that protein truncating variants (PTVs) in this gene are not causative of FA. Recently, eight individuals who were not diagnosed with FA were found to harbor bi-allelic *FANCM* PTVs. Three individuals developed early-onset cancers including lymphoblastic leukemia and squamous cancers [2]. In addition, five carriers of homozygous *FANCM* PTVs were identified among females diagnosed with breast cancer, two of which were diagnosed with early onset disease [3]. Hence, it seems that bi-allelic *FANCM* PTVs cause cancer predisposition with a greater risk for early onset development.

In the last years, results from case-control studies have indicated that mono-allelic *FANCM* PTVs are breast cancer risk factors. The c.5101C > T (p.Gln1701*, rs147021911) is frequent in the Finnish population with carrier frequency = 0.0162 versus 0.00168 in non-Finnish Europeans (NFE), as reported in gnomAD v2.1.1 (https://gnomad.broadinstitute.org/; [4]). In the Finnish population, p.Gln1701* was found to be associated with breast cancer risk with an odds ratio (OR) of 1.86 and with greater effects in familial cases, and for estrogen receptor-negative (ER-negative) and triple-negative breast cancer (TNBC) subtypes [5]. This variant was later found to be also associated with breast cancer survival and treatment outcome [6]. A second *FANCM* PTV, the c.5791C > T (p.Gly1906Alafs12*, rs144567652), which is annotated and hereafter referred to as p.Arg1931*, showed association with breast cancer risk in familial cases with OR = 3.93 [7]. This PTV was later found to also be associated with risk for TNBC in Finnish cases with OR = 5.14 [8]. A sequencing analysis of the entire *FANCM* coding region showed an excess of a third common PTV, the c.1972C > T (p.Arg658*, rs368728266), in German breast cancer cases versus controls, and confirmed that *FANCM* PTVs have a particularly high risk (OR = 3.75) for TNBC [9]. We recently tested p.Arg658*, p.Gln1701*, and p.Arg1931* in 67,112 European breast cancer cases and 53,766 controls. In these analyses, we observed that p.Arg658* was associated with increased risk of ER-negative disease and TNBC with ORs of  2.44 and  3.79, respectively, and that p.Arg1931* was associated with the risk of ER-negative breast cancer with an OR of 1.96 [10].

*BRCA1*, *BRCA2*, and *PALB2* are established breast cancer predisposition genes conferring high risk especially for ER-negative disease and TNBC. Similarly to other genes such as *BARD1*, *RAD51D*, *BRIP1*, and *RAD51C* [11], *FANCM* is emerging as a breast cancer predisposing factor conferring a greater risk specifically for these breast cancer subtypes. However, the spectrum of *FANCM* PTVs in Europe has never been investigated. Through a project call addressed to ENIGMA consortium collaborators [12], we collected data from European female breast cancer probands who were subjected to the sequencing of the *FANCM* coding region, resulting as carriers of a *FANCM* PTV. This study describes the spectrum of *FANCM* PTVs in this cohort of European breast cancer cases.

## 2. Results and Discussion

The 114 European breast cancer probands included in this study carried 27 different *FANCM* PTVs. Four, namely c.1491dupA (p.Gln498Thrfs*7, rs797045117), p.Arg658*, p.Gln1701*, and p.Arg1931*, were each identified in at least 8 probands and were classified as “common *FANCM* PTVs”. The remaining 23 variants were unique or were found in not more than three probands and were classified as “rare *FANCM* PTVs”. Additional data on common and rare *FANCM* PTVs were derived from previously published studies based on *FANCM* sequencing of German breast cancer probands [9,13]. Altogether, these and our data allowed us to show the geographic distribution of the four common PTVs and the rare PTVs combined in a cohort of breast cancer probands (Figure 1).

Of the 114 breast cancer probands, 86 (75.4%) carried one of the four common *FANCM* PTVs (Table 1). Of these 86 probands, 8 (9.3%), carried the p.Gln498Thrfs*7, 13 (15.1%) the p.Arg658*, 38 (44.2%) the p.Gln1701*, and 27 (31.4%) carried the p.Arg1931*. We observed that the relative frequency of the two most common PTVs, p.Gln1701* and p.Arg1931*, differ with respect to the country of origin of carriers. The p.Gln1701* is the most common PTV in Northern Europe while it has a lower relative frequency in Southern Europe. An opposite frequency gradient along the North–South axis appeared to exist for p.Arg1931*, which is the most common PTV in Southern Europe but more rare in Northern Europe. Consistently, in Central Europe, p.Gln1701* and p.Arg1931* are both common with similar relative frequencies. The relative frequencies of the *FANCM*:p.Arg658*, the third most common *FANCM* PTV, suggest that this variant has the highest frequency in Central Europe, is present in Southern Europe, and seems to be absent or very rare in Scandinavia. The fourth common *FANCM* PTV p.Gln498Thrfs*7 was identified in 6 of the 7 probands from Lithuania, suggesting it is a founder variant from this specific geographic area (Table 1, Figure 1).

The high relative frequency of *FANCM*:p.Gln498Thrfs*7 observed in Lithuania could indicate a founder effect in this relatively homogenous Baltic population. Similarly, haplotype analyses previously conducted in this population found a strong founder effect for several recurrent pathogenic variants in the high-risk predisposition genes *BRCA1* and *BRCA2* [14,15]. Calculations from haplotype analysis and paleo-demographic data indicate that a bottleneck phenomenon occurring approximately in the 4th–5th century AD in the Lithuanian region could have contributed to decreased genetic diversity and subsequent expansion of specific alleles [14].

We compared the geographic distribution of the four common PTVs observed in our study cohort with that reported in the gnomAD database v2.1.1 [4]. In gnomAD, 630 European individuals carried one of the p.Gln498Thrfs*7, p.Arg658*, p.Gln1701* or p.Arg1931* PTVs with relative proportions of 3.2% (20/630), 3.3% (21/630), 54.0% (340/630), and 39.5% (249/630), respectively. About half of the p.Gln498Thrfs*7 gnomAD carriers (9/20) were from Estonia, confirming that this variant has a specific origin in the Baltic countries. In gnomAD, p.Arg658* was prevalently found in NFE with the higher frequency of the carries (8/21) found in North-western Europeans and only one carrier found among Southern Europeans, which is similar to what we observed. Having being found in 340 gnomAD individuals, *FANCM*:p.Gln1701* is by far the most common PTV in Europe. As mentioned before, p.Gln1701* was particularly frequent in Finland and was observed more frequently in North-western than Southern Europeans, which is consistent with our data. *FANCM*:p.Arg1931* was the second most common *FANCM* PTV in Europe and the most common *FANCM* PTV in Southern Europeans in the gnomAD dataset. However, this variant was reported to be the most common also in North-western Europeans, not confirming the South–North frequency gradient observed in our cohort.

Of the 114 breast cancer probands, 29 (25.4%) carried one of the 23 rare *FANCM* PTVs (Table 2). Interestingly, a proband from the Czech Republic was a bi-allelic carrier of *FANCM* PTVs having inherited the rare p.Arg1030* variant from the mother, and the common p.Arg658* variant from the paternal side of the family (Table 1 and Table 2). This carrier was diagnosed with breast cancer at a young age (28 years), which is consistent with published data showing that females with bi-allelic *FANCM* PTV may develop early onset disease [3]. Eight of the 23 rare PTVs were annotated in the public ClinVar (https://www.ncbi.nlm.nih.gov/clinvar/), LOVD (https://www.lovd.nl/), or gnomAD databases and 15 were novel. Of the 23 rare PTVs, 19 were unique; two, p.Arg754* and p.Glu774*, were found in three probands; and two, p.Lys1615* and p.Thr1923Profs*2, in two probands. Given these low frequencies, we could not speculate on the origins or the geographic distribution of any of the rare *FANCM* PTVs. Considering the numbers of carriers of common PTVs and rare PTVs in each country, we observed that in Spain and France, there was an apparent excess of carriers of rare PTVs. On the contrary, in Sweden and Lithuania, no carriers of rare PTVs were found (Figure 1).

## 3. Materials and Methods

The individuals included in our analysis were contributed by ENIGMA consortium collaborators [12] interested in studying the effect/role of *FANCM* on breast cancer predisposition. They were originally ascertained at several Centers or National Studies and considered eligible for mutation sequencing analysis of breast and/or ovarian cancer predisposing genes in the frame of diagnostic or research purposes. In general, the ascertainment criteria for mutation testing were based on the presence of family history for breast or ovarian cancer, and on the early onset or specific clinical subtype of breast cancer. In the present study, we included 114 probands who received (i) a diagnosis of breast cancer, (ii) tested negative for *BRCA1* or *BRCA2* pathogenic variants, (iii) underwent complete sequencing of the *FANCM* coding region, resulting as a carrier of a PTV. Unfortunately, it was not possible to retrieve the number of probands that were originally subjected to mutation sequencing analysis at each participating Center or National Study. The 114 carriers of *FANCM* PTVs were ascertained in 20 Centers or National Studies from 13 European countries (Appendix A) including Czech Republic (20, 17.5%), Denmark (4, 3.5%), Finland (4, 3.5%), France (15, 13.2%), Greece (6, 5.3%), Hungary (2, 1.8%), Italy (3, 2.6%), Lithuania (7, 6.1%), Macedonia (2, 1.8%), Poland (3, 2.6%), Portugal (3, 2.6%), Spain (15, 11.4%), and Sweden (32, 28.1%). Some of these *FANCM* PTVs carriers were described in studies previously published, as indicated (Appendix A). Of the 114 probands, 58 (50.9%) were ER-positive, 27 (23.7%) were ER-negative, and 80 (70.2%) had a positive family history of breast or ovarian cancer. The mean age at breast cancer diagnosis was 46.7 years (min = 21, max = 85). All the 114 probands provided written informed consent for their data to be used for research purposes or specifically agreed to be tested for new variants in potential breast cancer associated genes. The participation of all the Centers or National Studies in the present work (Appendix A) was approved by their ethic committees. Specifically, these are the Ethics Committee of the General University Hospital in Prague (Approval code 87/14); Ethics Committee of Carlos III Institute of Health in Madrid (CEI PI 06_2013-v2); by the appropriate Advisory Committee on the Treatment of Health Research Information (Comité Consultatif de Protection des Personnes dans la Recherche Biomédicale (CCPPRB) Ile-de-France III) and by the National Data Protection authority; Ethics Committee of Fondazione IRCCS Istituto Nazionale dei Tumori in Milan (INT 7/15); Ethics Committee of the Capital Region of Denmark (H-4-2010-050); Ethics Committee of Hospital District of Helsinki and Uusimaa (HUS/1597/2016); Ethical Sub-Committee on Medicine, Pharmacy, Veterinary and Dental Medicine, Macedonian Academy of Sciences and Arts (09-1047/1 from 20/04/2016); IPO (Instituto Português de Oncologia) Porto institutional review board (nr. 219-019); Comité Territorial de Ética de la Investigación de Santiago -Lugo, Conselleria de Sanidade, Xunta de Galicia (2018/200); Ethics Committee of the Area Vasta Romagna e I.R.S.T (CEAV, 1727/2012 I.5/43); Provincial Ethics Committee (209/16) and the Modena Hospital (3387); Ethics Committee of University of Rzeszow, Poland (resolution no. 9/06/2015), University of Melbourne Human Research Ethics Committee, Melbourne, Australia; National Scientific and Ethics Committee (ETT-TUKEB), 20998-0/2010-1018EKU (845/PI/010); IDIBELL (Institut d’Investigacio Biomedica de Bellvitge, Barcelona) Ethics Committee PR278/19; Lithuanian Bioethics Committee (LT-HCCA-003 protocol), Vilnius Regional Ethics Committee (Nr. 58200-12-27(1-61); Ethics Committee of Lund, Sweden (Dnr 2011/349 and 2011/652); Bioethics Committee of NSCR (National Centre of Scientific Research) Demokritos (BcncSrD-240/eHΔ/11.3, updated on 29 June 2015).

## 4. Conclusions

We described the geographic distribution and relative prevalence of 27 different *FANCM* PTVs detected in 114 female European breast cancer probands with no pathogenic variants in *BRCA1* or *BRCA2* genes. As far as we know, this is the largest available collection of breast cancer probands carrying *FANCM* PTVs. In our study-cohort, we found that the *FANCM*: p.Gln1701* had high relative frequency in Northern Europe and low relative frequency in Southern Europe, whereas p.Arg1931* was the most common PTV in Southern Europe and less common in Northern Europe. The p.Arg658* had higher frequency in Central Europe, while p.Gln498Thrfs*7 is probably a founder variant from Lithuania. Fifteen of the 27 PTVs described are novel, including one found in three probands, and one found in two probands. Further data are warranted to provide an extensive spectrum of *FANCM* PTVs in Europe.

## Figures and Tables

**Figure 1 cancers-12-00292-f001:**
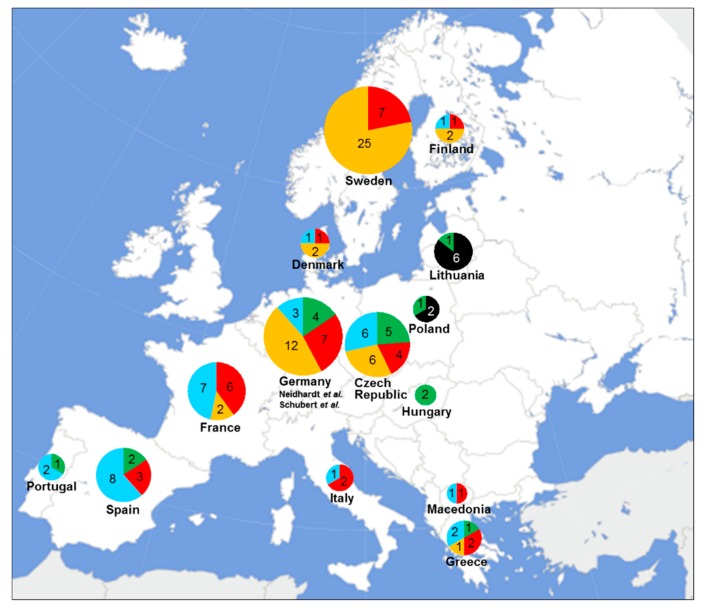
Distribution of the common and rare *FANCM* PTVs found in the 114 European female breast cancer probands. The common PTVs p.Gln498Thrfs*7, p.Arg658*, p.Gln1701*, and p.Arg1931* are indicated in black, green, orange, and red, respectively; the rare PTVs are indicated in light blue. Numbers of PTV carriers are reported according to their country of origin. The 26 PTV carriers from Germany are derived from previously published studies (Neidhardt et al., 21 probands [9]; Schubert et al., 5 probands [13]).

**Table 1 cancers-12-00292-t001:** Eighty-six breast cancer probands who carry one of the four common *FANCM* PTVs.

Center/Study	Country of Origin	PTV	Age at BC Diagnosis	ER-Status	BC/OC Family History
CZECANCA-Charles University, Prague	Czech Republic	p.Arg658*	53	ER-pos	yes
CZECANCA-Charles University, Prague	Czech Republic	p.Arg658*	85	NA	no
CZECANCA-Charles University, Prague	Czech Republic	p.Arg658*	35	NA	yes
CZECANCA-Gennet, Prague	Czech Republic	p.Arg658*	66	ER-pos	yes
CZECANCA-GHC, Prague	Czech Republic	p.Arg658* (#)	28	ER-pos	yes
CZECANCA-Charles University, Prague	Czech Republic	p.Gln1701*	51	ER-neg	yes
CZECANCA-Charles University, Prague	Czech Republic	p.Gln1701*	36	ER-pos	no
CZECANCA-Gennet, Prague	Czech Republic	p.Gln1701*	40	ER-neg	no
CZECANCA-Gennet, Prague	Czech Republic	p.Gln1701*	43	NA	no
CZECANCA-GHC, Prague	Czech Republic	p.Gln1701*	45	ER-pos	yes
CZECANCA-GHC, Prague	Czech Republic	p.Gln1701*	44	ER-pos	no
CZECANCA-Charles University, Prague	Czech Republic	p.Arg1931*	41	ER-neg	yes
CZECANCA-Charles University, Prague	Czech Republic	p.Arg1931*	46	ER-pos	no
CZECANCA-Charles University, Prague	Czech Republic	p.Arg1931*	44	ER-pos	yes
CZECANCA-GHC, Prague	Czech Republic	p.Arg1931*	63	NA	no
Copenhagen Breast Cancer Study	Denmark	p.Gln1701*	58	ER-neg	yes
Copenhagen Breast Cancer Study	Denmark	p.Gln1701*	50	ER-pos	no
Copenhagen Breast Cancer Study	Denmark	p.Arg1931*	27	ER-pos	yes
Helsinki Breast Cancer Study	Finland	p.Gln1701*	46	ER-pos	yes
Helsinki Breast Cancer Study	Finland	p.Gln1701*	43	ER-pos	yes
Helsinki Breast Cancer Study	Finland	p.Arg1931*	64	ER-pos	yes
GENE SISters	France	p.Gln1701*	44	ER-pos	yes
Institut Curie, Paris	France	p.Gln1701*	63	NA	yes
GENE SISters	France	p.Arg1931*	54	ER-pos	yes
GENE SISters	France	p.Arg1931*	39	ER-pos	yes
GENE SISters	France	p.Arg1931*	65	ER-pos	yes
GENE SISters	France	p.Arg1931*	45	ER-pos	yes
GENE SISters	France	p.Arg1931*	43	NA	yes
Institut Curie, Paris	France	p.Arg1931*	48	NA	yes
National Center for Scientific Research-Demokritos, Athens	Greece	p.Arg658*	33	ER-neg	no
National Center For Scientific Research-Demokritos, Athens	Greece	p.Gln1701*	61	ER-pos	yes
National Center For Scientific Research-Demokritos, Athens	Greece	p.Arg1931*	63	ER-neg	yes
National Center For Scientific Research-Demokritos, Athens	Greece	p.Arg1931*	42, 45	ER-pos	no
Hungarian Breast and Ovarian Cancer Study	Hungary	p.Arg658*	44	ER-pos	yes
Hungarian Breast and Ovarian Cancer Study	Hungary	p.Arg658*	35	ER-pos	yes
Fondazione IRCCS-Istituto Nazionale dei Tumori, Milan	Italy	p.Arg1931*	49	NA	yes
Istituto Scientifico Romagnolo per lo Studio e la Cura dei Tumori, Meldola	Italy	p.Arg1931*	32	ER-pos	no
Vilnius University Hospital Santaros Klinikos	Lithuania	p.Gln498Thrfs*7	31	ER-neg	no
Vilnius University Hospital Santaros Klinikos	Lithuania	p.Gln498Thrfs*7	35	ER-neg	no
Vilnius University Hospital Santaros Klinikos	Lithuania	p.Gln498Thrfs*7	46	ER-pos	no
Vilnius University Hospital Santaros Klinikos	Lithuania	p.Gln498Thrfs*7	39	ER-pos	no
Vilnius University Hospital Santaros Klinikos	Lithuania	p.Gln498Thrfs*7	58	ER-pos	yes
Vilnius University Hospital Santaros Klinikos	Lithuania	p.Gln498Thrfs*7	75	ER-pos	no
Vilnius University Hospital Santaros Klinikos	Lithuania	p.Arg658*	39	ER-pos	no
Macedonian Breast Cancer Study	Macedonia	p.Arg1931*	65	ER-neg	no
Wroclaw Medical University, Wrocław	Poland	p.Gln498Thrfs*7	63	ER-neg	no
Wroclaw Medical University, Wrocław	Poland	p.Gln498Thrfs*7	52	NA	yes
Wroclaw Medical University, Wrocław	Poland	p.Arg658*	62	ER-pos	yes
Portuguese Oncology Institute-Porto Breast Cancer Study	Portugal	p.Arg658*	29	NA	yes
Fundación Pública Galega Medicina Xenómica, Santiago de Compostela	Spain	p.Arg658*	40	ER-neg	no
Spanish National Cancer Research Centre, Madrid	Spain	p.Arg658*	38	ER-pos	yes
Catalan Institute of Oncology, Barcelona	Spain	p.Arg1931*	48	ER-pos	yes
Catalan Institute of Oncology, Barcelona	Spain	p.Arg1931*	55	ER-pos	yes
Spanish National Cancer Research Centre, Madrid	Spain	p.Arg1931*	47	NA	yes
Swedish Breast Cancer Study	Sweden	p.Gln1701*	34	ER-neg	yes
Swedish Breast Cancer Study	Sweden	p.Gln1701*	58	ER-neg	yes
Swedish Breast Cancer Study	Sweden	p.Gln1701*	21	ER-neg	yes
Swedish Breast Cancer Study	Sweden	p.Gln1701*	71	ER-neg	yes
Swedish Breast Cancer Study	Sweden	p.Gln1701*	71	ER-neg	yes
Swedish Breast Cancer Study	Sweden	p.Gln1701*	57	ER-pos	yes
Swedish Breast Cancer Study	Sweden	p.Gln1701*	37	ER-pos	yes
Swedish Breast Cancer Study	Sweden	p.Gln1701*	68	ER-pos	yes
Swedish Breast Cancer Study	Sweden	p.Gln1701*	43	ER-pos	yes
Swedish Breast Cancer Study	Sweden	p.Gln1701*	54	ER-pos	yes
Swedish Breast Cancer Study	Sweden	p.Gln1701*	56	ER-pos	yes
Swedish Breast Cancer Study	Sweden	p.Gln1701*	48	ER-pos	yes
Swedish Breast Cancer Study	Sweden	p.Gln1701*	27	ER-pos	no
Swedish Breast Cancer Study	Sweden	p.Gln1701*	42	ER-pos	yes
Swedish Breast Cancer Study	Sweden	p.Gln1701*	60	ER-pos	yes
Swedish Breast Cancer Study	Sweden	p.Gln1701*	44	ER-pos	yes
Swedish Breast Cancer Study	Sweden	p.Gln1701*	57	ER-pos	yes
Swedish Breast Cancer Study	Sweden	p.Gln1701*	32	ER-pos	yes
Swedish Breast Cancer Study	Sweden	p.Gln1701*	46	NA	yes
Swedish Breast Cancer Study	Sweden	p.Gln1701*	49	NA	yes
Swedish Breast Cancer Study	Sweden	p.Gln1701*	36	NA	no
Swedish Breast Cancer Study	Sweden	p.Gln1701*	46	NA	yes
Swedish Breast Cancer Study	Sweden	p.Gln1701*	39	NA	no
Swedish Breast Cancer Study	Sweden	p.Gln1701*	28	NA	yes
Swedish Breast Cancer Study	Sweden	p.Gln1701*	47	NA	yes
Swedish Breast Cancer Study	Sweden	p.Arg1931*	36	ER-neg	yes
Swedish Breast Cancer Study	Sweden	p.Arg1931*	47	ER-neg	no
Swedish Breast Cancer Study	Sweden	p.Arg1931*	52	ER-pos	yes
Swedish Breast Cancer Study	Sweden	p.Arg1931*	55	ER-pos	yes
Swedish Breast Cancer Study	Sweden	p.Arg1931*	36	ER-pos	no
Swedish Breast Cancer Study	Sweden	p.Arg1931*	39	NA	yes
Swedish Breast Cancer Study	Sweden	p.Arg1931*	56	NA	yes

PTV, protein truncating variant; BC, breast cancer; OC, ovarian cancer; ER, estrogen receptor; pos, positive; neg, negative; NA, not available. (#) This individual also carries the rare *FANCM*: c.3088C > T (p.Arg1030*) PTV and is also included in Table 2.

**Table 2 cancers-12-00292-t002:** Twenty-nine breast cancer probands who carry one of the 23 rare *FANCM* PTVs.

Center/Study	Country of Origin	PTV	Annotated in Databases	Age at BC Diagnosis	ER-Status	BC/OC Family History
CZECANCA-AGEL, Novy Jicin	Czech Republic	c.1798C > T; p.Gln600*	no	37	ER-pos	yes
CZECANCA-Charles University, Prague	Czech Republic	c.3898G > T; p.Glu1300*	no	37	ER-pos	no
CZECANCA-Charles University, Prague	Czech Republic	c.3979_3980delCA; p.Gln1327Valfs*16	LOVD, gnomAD	50	ER-pos	yes
CZECANCA-Gennet, Prague	Czech Republic	c.2260C > T; p.Arg754*	gnomAD	51	ER-neg	no
CZECANCA-Gennet, Prague	Czech Republic	c.2260C > T; p.Arg754*	gnomAD	34	ER-neg	no
CZECANCA-GHC, Prague	Czech Republic	c.3088C > T; p.Arg1030* (#)	no	28	ER-pos	yes
Copenhagen Breast Cancer Study	Denmark	c.3745_3748del; p.Thr1249Glnfs*7	no	50	ER-pos	NA
Helsinki Breast Cancer Study	Finland	c.4021_4022del	no	39	ER-pos	yes
GENE SISters	France	c.1432C > T; p.Arg478*	no	48	NA	yes
GENE SISters	France	c.3745dupA; p.His1248fs	no	63	NA	yes
Institut Curie, Paris	France	c.1868delA; p.Arg624Glufs*46	no	56	ER-neg	yes
Institut Curie, Paris	France	c.2260C > T; p.Arg754*	gnomAD	43	ER-neg	yes
Institut Curie, Paris	France	c.2590delG; p.Asp864Ilefs*12	no	64	ER-neg	yes
Institut Curie, Paris	France	c.4930C > T; p.Arg1644*	gnomAD	46	NA	yes
Institut Curie, Paris	France	c.5363delCTAG; p.Ser1788Leufs*15	no	43	ER-neg	no
National Center For Scientific Research-Demokritos, Athens	Greece	c.4843A > T; p.Lys1615*	LOVD	33	NA	yes
National Center For Scientific Research-Demokritos, Athens	Greece	c.5314_5315delTG; p.Cys1772*	LOVD, gnomAD	47	ER-pos	yes
Fondazione IRCCS-Istituto Nazionale dei Tumori, Milan	Italy	c.2354T > G; p.Leu785*	no	35	NA	no
Macedonian Breast Cancer Study	Macedonia	c.5048_5052delAAAGA; p.Lys1683Argfs*3	gnomAD	32	ER-neg	no
Portuguese Oncology Institute-Porto Breast Cancer Study	Portugal	c.2320G > T; p.Glu774*	no	42	ER-neg	yes
Portuguese Oncology Institute-Porto Breast Cancer Study	Portugal	c.5766_5769del; p.Thr1923Profs*2	no	43	ER-neg	yes
Catalan Institute of Oncology, Barcelona	Spain	c.624_625delAA; p.Ile208fs	no	23	NA	no
Catalan Institute of Oncology, Barcelona	Spain	c.1303C > T; p.Gln435*	no	39	ER-pos	yes
Catalan Institute of Oncology, Barcelona	Spain	c.5766_5769del; p.Thr1923Profs*2	no	31	ER-pos	no
Fundación Pública Galega Medicina Xenómica, Santiago de Compostela	Spain	c.2320G > T; p.Glu774*	no	57	ER-neg	yes
Fundación Pública Galega Medicina Xenómica, Santiago de Compostela	Spain	c.2586_2589delAAAA; p.Lys863Ilefs*12	ClinVar, LOVD, gnomAD	49	NA	yes
Spanish National Cancer Research Centre, Madrid	Spain	c.1196C > G; p.Ser399*	ClinVar	66	NA	yes
Spanish National Cancer Research Centre, Madrid	Spain	c.2320G > T; p.Glu774*	no	40, 65	ER-pos	yes
Spanish National Cancer Research Centre, Madrid	Spain	c.4843A > T; p.Lys1615*	LOVD	41	NA	yes

PTV, protein truncating variant; BC, breast cancer; OC, ovarian cancer; ER, estrogen receptor; pos, positive; neg, negative; NA, not available. (#) This individual also carries the common PTV, p.Arg658*, and is also included in Table 1.

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
