# Peer review of "The Spectrum of FANCM Protein Truncating Variants in European Breast Cancer Cases"

_cancers, 2020, doi:10.3390/cancers12020292_

Round 1
Reviewer 1 Report
This study was aimed to investigate the distribution and frequency of FANCM germline protein truncating variants in European breast cancer cases. Geographical distribution and relative prevalence of four common and 23 rare FANCM PTVs including 15 novel variants is described. Although a number of patients in a studied cohort is not so large (116 breast cancer cases), this is the largest available collection of breast cancer patients carring this type of analysis sufficient to be published. Data from this analysis could be verified in a larger sets of patients in the future. A subscribed manuscript is capable of being published depending on minor revision process.
Minor points:
Introduction; some other genes associated with higher risk of TNBC such as BRCA1/2, PALB2 should be mentioned for comprehensive information.
Materials and Methods; Authors should include brief information or reference describing the exact methodology of sequencing used in all studies presented in Table 1.
Author Response
Introduction; some other genes associated with higher risk of TNBC such as BRCA1/2, PALB2 should be mentioned for comprehensive information.
- Response. As suggested BRCA1, BRCA2 and PALB2 were mentioned in the last paragraph of the introduction.
Materials and Methods; Authors should include brief information or reference describing the exact methodology of sequencing used in all studies presented in Table 1.
- Response. A description of the methodology of sequencing used in all studies presented in now included in Table S1
Reviewer 2 Report
In this report, Figlioli G. and colleagues describe the European geographic distribution of FANCM PTVs in a large group of breast cancer cases.
I would suggest the authors to briefly correlate genetic data and clinical pathological features of probands in order to shed light on the peculiar phenotype of FANCM PTVs carriers and to enforce the hypothesis that these variants are pathogenic.
Author Response
I would suggest the authors to briefly correlate genetic data and clinical pathological features of probands in order to shed light on the peculiar phenotype of FANCM PTVs carriers and to enforce the hypothesis that these variants are pathogenic.
- Response. While this would certainly be of interest we need to note that in the scientific literature there are many articles showing that FANCM PTVs are associated with breast cancer risk, with greater effect for ER-negative or TNBC subtypes. These articles were mentioned in the introduction section of the present manuscript. The specific aim of the present manuscript was to describe the geographic distribution and relative prevalence of FANCM PTVs in a European cohort exclusively selected among breast cancer cases. Hence, we cannot provide data enforcing that these variants are pathogenic nor that they are specifically associated with breast cancer risk. Nevertheless, we described a bi-allelic carrier of FANCM PTVs who, as reported in the Results and Discussion was diagnosed with breast cancer at a young age (28 years). This is consistent with previously published data showing that females with bi-allelic FANCM PTV may develop early onset disease (Catucci et al., Genet Med 2018).
Reviewer 3 Report
This investigation, titled with “The spectrum of FANCM protein truncating variants in European breast cancer cases”, deserves for publication in the journal “Cancers” in that the authors provide scientific community and the world with valuable spectrum and distribution of FANCM protein truncating mutations associated with breast carcinogenesis.
One comment I am going to give the authors is that it is better to present protein domain insight on how each kind of FANCM protein truncating mutation may remove functional domains by using Pfam domain database.
Author Response
One comment I am going to give the authors is that it is better to present protein domain insight on how each kind of FANCM protein truncating mutation may remove functional domains by using Pfam domain database.
- Response. While this would certainly be of interest, we need to note that the aim of our study was to provide the spectrum distribution of the FANCM PTVs in Europe and not to infer or comment on the effect of each PTVs on the protein. While each single FANCM PTV could cause a different damage to the protein, or confer different risk, for the purpose of the present study all the FAMCM PTVs are considered breast cancer risk factors.